# Migration of Polypropylene Oligomers into Ready-to-Eat Vegetable Soups

**DOI:** 10.3390/foods9101365

**Published:** 2020-09-25

**Authors:** Chiara Conchione, Paolo Lucci, Sabrina Moret

**Affiliations:** Department of Agri-Food, Environmental and Animal Sciences, University of Udine, 33100 Udine, Italy; paolo.lucci@uniud.it (P.L.); sabrina.moret@uniud.it (S.M.)

**Keywords:** polyolefin oligomeric hydrocarbons (POH), polypropylene (PP), migration, microwave heating, on-line HPLC-GC

## Abstract

Polyolefin oligomeric hydrocarbons (POH) are non-intentionally added substances (NIAS) which mainly reside in the polymer (PE, PP) as a consequence of the polymerization process, and that under favorable conditions (high fat content, high temperature, and long contact time) may migrate at high amount from the packaging into the food. The food industry offers a wide range of ready-to-eat products, among these, vegetable soups designed to be stored at refrigeration temperature (for times around 6 weeks), and in most cases to be heated for a few minutes in a microwave oven (into the original container, mostly of PP) before consumption. The present work aimed to study for the first-time migration of POH during the shelf life of these products, including storage at refrigeration temperature and after microwave heating. On-line high-performance liquid chromatography (HPLC)-gas chromatography (GC), followed by flame ionization detection (FID), was applied for POH analysis in a number of ready-to-eat products purchased from the Italian market. Microwave heating determined a variable POH increase ranging from 0.1 to 6.2 mg/kg. Parameters possibly affecting migration such as fat content and heating time were also studied.

## 1. Introduction

Polyolefins such as PE and PP represent the food packaging materials most widely used in direct contact with food. Among the potential migrants of polyolefins [1], oligomers, also known as polyolefin oligomeric hydrocarbons (POH), are one of the less-studied classes of migrants.

Oligomers, that for definition consist of a finite number of repeating units, can be formed during the polymerization reaction between monomers (side reaction products such as cyclic oligomers can be formed) or might residue due to incomplete polymerization [2]. Some authors demonstrated that POH are not formed during other processing steps (film extrusion, corona treatment, blow molding) [3].

Oligomers can be also used as reacting blocks (prepolymers) for polymer manufacture. According to their origin, oligomers are considered non intentionally added substances (NIAS), but they can be also added intentionally, to provide important properties to the polymer [2].

Both intentionally and non-intentionally added oligomers (below 1000 Da) need an adequate risk assessment and are subject to Commission Regulation No 10/2011 (Art. 19), and materials containing them must be compliant with Art. 3 of the Framework Regulation No 1935/2004.

Due to the limited resorption of larger molecules, only oligomers with a molecular weight (MW) of less than 1000 Da are of concern and need an adequate risk assessment. According to Barp et al. [4], due to their similarity to mineral oil saturated hydrocarbons (MOSH), POH presence in food is of concern and their safety should be evaluated considering their bioaccumulation potential in human tissues. On the other hand, a project commissioned by PlasticsEurope to the Fraunhofer Institute concluded that, if the maximum allowed overall migration limit of 60 mg/kg food is respected, oligomer migration from the plastics to food, and consequent dietary exposure, are below any toxicological level of concern [5].

POH comprises saturated hydrocarbons (POSH), and variable amounts of monounsaturated hydrocarbons (POMH). POSH from different polyolefins (LDPE, HDPE, PP, etc.) have characteristic chromatographic profiles, which allow one to distinguish the different materials: they mainly consist of linear and branched alkanes with unique patterns. For example, the presence of linear and branched cyclopentane and cyclohexane has been observed in PE, but not in PP [6]. Reaction products (i.e., oxidation products) can be also present in the polymers [7].

On-line HPLC-GC, according to the method developed by Biedermann et al. [8], allows for the rapid determination of POH, but not for distinguishing them from MOSH when they are both present and MW distributions are very close to each other. For a more reliable determination, Biedermann end co-authors proposed to combine LC pre-separation to GC × GC [9], which represents the most powerful technique to characterize complex hydrocarbon mixtures. A rather complex HPLC-HPLC-GC-FID system with a first silica gel column to separate MOSH and MOAH, and a second silver-impregnated HPLC column to achieve POSH and POMH separation, was proposed by Lommatzsch et al. [10].

Even though under favorable conditions POH may migrate from the packaging into the food, till now a few works investigated on this subject. As demonstrated by Lommatzsch [2], who determined oligomer migration from the known sealing layer into food simulants (sunflower oil) and real food subjected to pasteurization with different fat content (tomato, carbonara, and pesto sauces), high-fat content, and high contact temperatures are expected to accelerate migration.

Some authors quantified POH migration into various dry and wet foods, including infant formula, using on-line HPLC-GC-FID without POSH and POMH separation [11,12]. In a similar manner, Barp and co-author analyzed POH in semolina and egg pasta packaged in PP film, finding migration up to 0.6 and 1.7 mg/kg, respectively [13].

The food industry offers a wide range of ready-to-eat products, among these vegetable soups (and other products), designed to be stored at refrigeration temperature for several weeks and to be heated in the packaging container before use, mostly by applying microwave heating.

The present work aimed to evaluate, for the first time, POH residues in PP containers used in contact with vegetable soups, and to evaluate migration into the food product after microwave heating, according to the instruction reported on the label. The influence of parameters possibly affecting migration, such as heating time and the fat amount was also studied, as well as migration behavior in food simulants.

## 2. Materials and Methods

### 2.1. Reagents and Standards

All solvents were from Sigma-Aldrich (Milan, Italy). Hexane and dichloromethane were distilled before use. Ethanol was of HPLC grade. Water was purified with a Milli-Q system (Millipore, Bedford, MA, USA).

The *n*-alkane C_10_-C_40_ standard mixture (50 mg/L each) used for assessing GC performance was purchased from Sigma-Aldrich. Internal standards were from Supelco (Milan, Italy). The working standard contained: 5-α-cholestane (Cho, 0.6 mg/mL), *n*-C_11_ (0.3 mg/mL), *n*-C_13_ (0.15 mg/mL), cyclohexylcyclohexane (CyCy, 0.3 mg/mL), *n*-pentylbenzene (5B, 0.30 mg/mL), 1-methylnaphthalene (1-MN, 0.30 mg/mL), 2-methylnaphthalene (2-MN, 0.30 mg/mL), tritert-butylbenzene (TBB, 0.3 mg/mL), and perylene (Per, 0.6 mg/mL) in toluene.

Before use, all the glassware was carefully washed and rinsed with distilled acetone and hexane.

### 2.2. Instrumentation

The microwave extractor used for POH extraction from food products was a Mars-X (CEM Corporation, Matthews, NC, USA) able to process up to 14 samples simultaneously.

The domestic microwave oven was a Whirlpool model IT369 (Whirpool Corporation, Benton Harbor, MI, USA).

The on-line HPLC–GC instrument was an LC–GC 9000 from Brechbühler (Zurich, Switzerland) and consisted of a Phoenix 40 with three syringe LC pumps and four switching valves, and a UV/Vis detector (UV-2070 Plus; Jasco, Tokyo, Japan). The GC was a Trace GC Ultra from Thermo Scientific (Milan, Italy). The autosampler was a PAL LHS2-xt Combi PAL (CTC, Zwingen, Switzerland).

The concentration unit consisted of a centrifuge (Univapo 100 H, Uniequip System; Martinsrieder, Munich, Germany), connected to a vacuum pump (Buchi, Flawil, Switzerland).

### 2.3. Samples

A number of vegetable soup samples plus 1 potato puree (11 product types of 6 different brands) were analyzed for their POH content, both before and after microwave heating. Respective PP containers were also analyzed before microwave heating. Heating conditions were the most drastic of those indicated on the label and reported in Table 1. Most of the vegetable soups contained only vegetables (V), while others contained also legumes (L) and/or cereals (C). All samples were directly purchased from the supermarket. To evaluate variability due to different production lot, in one case 3 samples of the same product (V4), but different production lot were purchased and compared with two samples of the same product (VLC) and same lot.

Table 1 reports a description of the samples, suggested heating conditions, indications on the brand type, container weight, food weight, and fat content. Instruction on the label indicated to drill the protective film (or to lift it before heating). Only in one case, it was indicated to remove the protective layer before heating. Suggested heating conditions were 3 or 4 min at a power ranging from 750 to 1000 W. Containers were all made of PP.

### 2.4. Extraction

Oligomer extraction from the plastic container was performed according to the method proposed by Biedermann-Brem et al. [11] and using microwave extraction. As the container before the contact with the food was not available, a strip of the plastic material, taken half a centimeter below the upper edge of the container (the part not in contact with the food) was cut into little pieces. About 300 mg were exactly weighted into an extraction vial, added with 20 μL of the internal standard, 10 mL of hexane and extracted overnight under magnetic stirring, or added with 20 mL of a mixture hexane/ethanol 1:1 (*v*/*v*) and microwave extracted for 30 min at 80 °C. After cooling and water addition, the hexane phase was recovered. To verify extraction exhaustiveness, preliminary trials under different temperature, extraction time, and heating mode (traditional oven vs. microwave oven) were carried out.

Food samples were extracted according to the method described by Moret et al. [14]. Briefly, 5 g of sample, previously homogenized (if necessary), were directly weighted into a Teflon-lined vessel, added with 5 μL of internal standard solution, 10 mL of saturated methanolic potassium hydroxide (KOH), and 10 mL of *n*-hexane. Microwave-assisted saponification (MAS) and simultaneous unsaponifiable extraction were carried out at 120 °C for 20 min. After cooling the vessels were opened and added with about 20 mL of water and 3–4 mL of methanol (without mixing) and left to rest for about 20 min at −20 °C to facilitate phase separation avoiding emulsion formation. After concentration (5 mL were concentrated to 1 mL), the hexane extract underwent an on-line HPLC-GC analysis. Recovery tests were performed on a home-made pureed zucchini spiked with a known amount of POH previously extracted from a PP container, while repeatability was assessed on a real sample that underwent the entire procedure six times.

### 2.5. Chromatographic Conditions for HPLC-GC Analysis

The HPLC column was a 25 cm × 2.1 mm i.d Lichrospher Si 60, 5 μm (DGB, Schlossboeckelheim, Germany). The GC was a Trace GC Ultra from Thermo Scientific (Milan, Italy).

A gradient, starting with hexane 100% and reaching 30% of dichloromethane after 0.5 min at a flow rate of 300 μL/min was used to separate the MOSH (fraction from 2.0 to 3.5 min) and the MOAH (fraction from 3.8 to 5.3 min). HPLC-GC transfer occurred through the Y-interface based on the retention gap technique and partially concurrent eluent evaporation [15]. A 10 m × 0.53 mm i.d. uncoated, deactivated precolumn was connected by a steel T-piece union to the solvent vapor exit (SVE) and a 15 m × 0.25 mm i.d. separation column coated with a 0.15 μm film of PS-255 (1% vinyl, 99% methylpolysiloxane) (Mega, Legnano, Italy). A rapid oven gradient (40 °C/min) starting from 55 up to 350 °C was used for GC analysis [16]. Hydrogen was used as the carrier gas with a constant pressure of 60 kPa increased at 90 kPa during the transfer of the fractions from LC to GC.

The FID and the SVE were heated at 360 and 140 °C, respectively. After the transfer, the LC column was backflushed with dichloromethane and reconditioned prior to the subsequent injection. The data were acquired and processed by the ExaChrom software (Brechbühler, Switzerland). Quantification was based on internal standards following the formula:(1)$$w_{hc}=\frac{A_{s\;}\times m_{is}\times 1000}{A_{is}\times m}$$
where:*w_hc_* is the mass fraction of the MOSH/POH in mg/kg*A_s_* is the area of the sample*A_is_* is the area of the internal standard peak*m_is_* is the mass, of the internal standard solution added to the sample, in mg*m* is the mass of the test portion, in g

When MOSH internal standard peaks coeluted with the POH, the MOAH standard or, the external standard calculation was used.

## 3. Results

### 3.1. POH Extraction from PP Containers

The first part of this work aimed to quantify POH in the plastic containers. To achieve this goal, extraction conditions were optimized to obtain complete recovery of hydrocarbons up to C_35_, limiting the extraction of higher molecular weight hydrocarbons, which tend to accumulate in the retention gap and the first part of the capillary column, determining a rapid decrease of the GC performance.

To check the completeness of POH extraction from PP containers, some preliminary trials were carried out on different aliquots of the same plastic material, which was in part reduced into squares of about 0.5 cm^2^, and in part cut into small particles of about 1 mm of diameter. Different aliquots (300 mg) of the same sample were extracted at ambient temperature (25 °C) for different times (overnight for 16 h, and for 72 h) with 10 mL of *n*-hexane. No differences were found between the samples cut into squares and those reduce to little particles, as well as between those extracted with hexane overnight (16 h) and those extracted with hexane for 72 h at ambient temperature. By increasing the extraction temperature, oligomers of higher molecular weight were extracted in a higher amount. With respect to extraction at 25 °C, extraction at 60 °C for 72 h allowed us to obtain comparable extraction of hydrocarbons up to C_21_, and slightly higher yield of higher molecular weight hydrocarbons (Figure 1) in the range up to C_35_. Particularly, the recovery of POH up to C_35_ at 25 °C were 94% of those obtained at 60 °C (72 h). Extraction at a higher temperature (80 °C for 30 min) using both traditional and microwave heating (hexane/ethanol 1:1 *v*/*v* as an extraction solvent in the latter case), gave, for POH up to C_35_, extraction yields comparable to those obtained at 60 °C for 72 h. Completeness of the extraction was further demonstrated with a second extraction at a higher temperature (120 °C) that determined the release of higher molecular weight oligomers, which made the retention gap and the GC column dirty meaning strong washing was necessary.

It is interesting to observe that microwave extraction at 80 °C for 30 min with hexane/ethanol 1:1 (*v*/*v*) allowed to obtain practically quantitative extraction of POH up to C_35_. Nevertheless, extraction at ambient temperature with hexane overnight gave only a slightly lower extraction yield in the range C_20_–C_35_ (overall yield over 93%). Repeatability trials (6 replicates of the same sample) gave a coefficient of variations lower than 4%, for overnight extraction, and lower than 6% for microwave-assisted extraction.

Based on these results, and considering acceptable the recovery obtained, extraction was performed by using overnight extraction at 25 °C.

### 3.2. POH Extraction from Vegetable Soups

Concerning POH extraction from the food, a home-made pureed zucchini with no POH and very low MOSH (less than 0.2 mg/kg), added with extra virgin olive oil (2%), was spiked with a known amount (around 6 mg/kg) of POH mixture extracted from a PP container. Four aliquots of the spiked sample underwent the entire procedure (MAS followed by on-line HPLC-GC-FID analysis). POH recovery calculated by comparing the chromatographic area of the spiked sample, subtracted from the contribution of the unspiked sample, with that of the added POH (previously extracted from a PP container and standardized to a known concentration), were practically quantitative (on average 102%) with RSD lower than 12%. Figure 2 shows the LC-GC chromatograms of the unspiked home-made pureed zucchini, of the added POH, and a replicate of the spiked samples.

Repeatability (6 replicates) was also tested on a real sample (V6), contaminated with 0.6 mg/kg of POH (RSD < 11%).

### 3.3. POH in Plastic Containers

Figure 3 reports POH amounts (mg/kg) found in the plastic containers, which for POH C_10–35_ ranged from 1096 to 4679 mg/kg. By considering the weight of the plastic container and the weight of the food in contact with the container, potential migration (which ranged from 53 to 242 mg/kg) was also calculated by assuming that all the POH present in the packaging migrated into the food during microwave heating.

Figure 3 reports also LC-GC-FID traces of two PP containers, the one with the highest contamination (4679 mg/kg of POH C_10–35_) and the one with the lowest contamination (1096 mg/kg of POH C_10–35_). All traces evidenced the presence of narrow humps of oligomers built of C_3_-unit, which are typical of PP.

It is interesting to observe that two samples with the lowest POH content (V5 and V6) were of the same plastic-type (same producer), indicating as, by choosing proper polymerization conditions, it is possible to obtain products with low residual POH. Except for these two samples, the other samples showed relatively low variability, with POH content varying of a factor of two. Three samples of the same product type (V4), but a different lot, showed higher variability (2011–3093 mg/kg) than two samples of the same product type and same lot (VLCa, 2375 mg/kg; VLCb, 2763 mg/kg).

According to Testoni and Mingozzi, the use of different catalysts in the polymerization process may lead to very different POH amounts [17]. Three PP samples obtained by the same polymerization plant, but different catalysts, showed residual oligomers ranging from 2450 to 6150 in the spheres.

### 3.4. POH/MOSH in Food Products before Microwave Heating

Table 2 reports POH/MOSH content (expressed in mg/kg) of the soup samples analyzed before microwave heating. Data obtained after microwave heating and net migration are also reported but will be discussed in the next paragraphs. Some samples were contaminated with MOSH. As POH cannot be clearly distinguished from MOSH by on-line HPLC-GC, results are expressed as POH/MOSH.

Most of the products contained detectable POH already before microwave heating, as well as some MOSH. Even though a precise quantification of the POH was not possible, it was roughly estimated from the HPLC-GC profile (most of the POH eluted in the range C_10–20_, generally free from MOSH contamination), POH amount ranging from around 0.2 to 2.0 mg/kg. POH/MOSH contamination before microwave heating ranged from 0.6 to 3.0 with an average of 1.4 mg/kg.

Figure 4 shows the HPLC-GC traces of a selection of samples with different contamination profiles before microwave heating.

Samples V4 had a profile characterized by the presence of little POH (well visible in the first part of the trace) and with MOSH (forming a hump) in the second part of the trace. Sample LP had a mixed MOSH/POH contamination with predominant MOSH, while sample V2 had a prevalent POH contamination around 2.0 mg/kg. V3 had the typical profile of a sample contaminated with polyalpha olefins (PAO), whose origin remained unknown as further investigations demonstrated that the contamination was not from the adhesive used to seal the cover.

To explain the presence of POH in the product before microwave heating, it was verified if some contamination could occur during storage a 4 °C. To this purpose, two soup samples (V4) were purchased in double (2 samples of the same lot) and analyzed before microwave heating (Figure 5): one just after being purchased at the supermarket and another one at the best before the end date (after a 4-week storage at 4 °C). Results showed little but detectable migration (less than 0.2 mg/kg) during storage at 4 °C. Migration before microwave heating is also expected during hot filling at the production plant, which is probably responsible for the relatively high POH content found in some soup samples before microwave heating.

### 3.5. MOSH/POH Content after Microwave Heating

All soup samples were in PP containers intended to be heated before consumption, some of which are of different types (different size, thickness, color). Microwave heating was performed according to the indications reported on the label (see Table 1). The final temperature after recommended microwave heating ranged from 60 to above 80 °C, also depending on the water content.

MOH/POH content of the vegetable soup samples after microwave heating is reported in Table 2. It ranged from 0.7 to 8.9 mg/kg (on average 3.3 mg/kg), which is about twice the average MOH/POH content found before microwave heating.

### 3.6. Net POH Migration Due to Microwave Heating

To calculate net POH migration, the results obtained for soup samples before microwave heating were subtracted from those obtained after the heating treatment.

Net POH increase varied considerably depending on the sample. It ranged from 0.1 to 6.2 (on average 2 mg/kg). Except for 4 samples (V3, V5, V6, and PU), which showed negligible POH increase (<0.4 mg/kg) after microwave heating, POH content increased considerably (from 4.0 to 6.2 mg/kg) for 3 samples (V1, V2, V4), and from 0.7 to 1.6 mg/kg for the other samples. As MOH are not affected by the microwave heating, the increase observed is all ascribable to POH. Figure 6 shows two examples: one where such an increase was well evident, and another where the increase was less evident, probably due to the lower POH amount in the plastic container.

Sample VC is a cereal soup with negligible MOSH contamination around (0.2 mg/kg). After microwave heating for 3 min at maximum power (heating condition reported on the label), POH content increased of a factor of 3. For another sample (V4), whose trace is not reported, POH content increased of a factor around 6.

By comparing actual POH migration with potential migration, the percentage of the potential was calculated. It was lower than 1% for samples V3, V6, LC, and PU, and comprised between 1% and 2% for samples V5, VLC, VC, and LP, and between 3% and 6% for other samples (V1, V2, and V4).

As demonstrated by these data, POH migrated in the food represented only a little part of what could potentially migrate from the PP container. This probably depends on the high-water content and low-fat content of these products. No clear correlation was found between the fat content and the amount of POH migrated, probably because of the influence of other parameters. The migration rate is probably influenced also by residual POH present in the container (in fact very low migration was observed for sample V5 and V6 which had the lowest POH content), sample composition, and by the different heating conditions used.

### 3.7. Effect of the Fat Content

To better investigate the effect of fat content on POH migration, a fresh pureed zucchini (prepared at home), was added with different amounts (1%, 2%, and 6%) of a clean extra virgin olive oil and homogenized with a Polytron until obtaining a homogeneous emulsion.

Aliquots (5 g each) of the pureed zucchini used as food simulant, were directly weighted in Teflon-lined extraction tubes, put in contact with 200 mg of plastic material, and heated in a domestic microwave oven at 700 W for a constant time (1 min) to maintain the same packaging to food weight ratio and reach temperature not higher than 80 °C, as occurring under real conditions. Figure 7 shows the effect of different oil amounts on POH migration.

Results obtained demonstrate that the fat content has an important impact, not only on the amount of POH migrated (the migration rate increased with increasing the oil content) but also on the molecular weight of the migrants. Migration of POH C_10–35_ increased from 0.6 mg/kg (1% of added oil) to 1.4 mg/kg (2% of added oil) and finally to 3.6 mg/kg (6% of added oil). At the highest oil addition, migration involved also a significant amount of POH beyond C_35_.

### 3.8. Effect of the Heating Time

To evaluate the effect of the heating time, 4 additional vegetable soups of the same type and same lot were subjected to microwave heating for 3, 4, 5, and 6 min. Net POH C_10–35_ migration after 3 min at 900 (recommended heating conditions) was relatively low (0.4 mg/kg), but increased with increasing the time (Figure 8), reaching levels of 1.4 mg/kg after 6 min.

## 4. Conclusions

PP containers, widely used to store and heat ready to use foods, can release POH. Fat content and microwave duration (at constant power) are important parameters affecting migration.

POH/MOSH levels in vegetable soups ranged from 0.6 to 3.0 mg/kg before microwave heating, and from 0.7 to 8.9 after microwave heating. Detectable POH amounts (well visible in the range C_10–20_ were MOSH interference is not evident) were already present in food before microwave heating, probably due to hot filling during the production. In general, POH content increased after microwave heating, but with respect to the potential migration, only a part of the POH was transferred to the food. The high variability found in residual POH content of plastic containers, and the amounts migrated in the food, leaves room to producers to look for improving their product to minimize POH migration in the final product.

## Figures and Tables

**Figure 1 foods-09-01365-f001:**
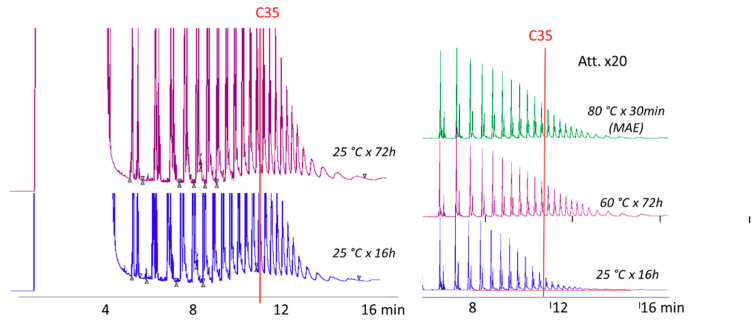
Optimization of polyolefin oligomeric hydrocarbons (POH) extraction from polypropylene (PP) containers.

**Figure 2 foods-09-01365-f002:**
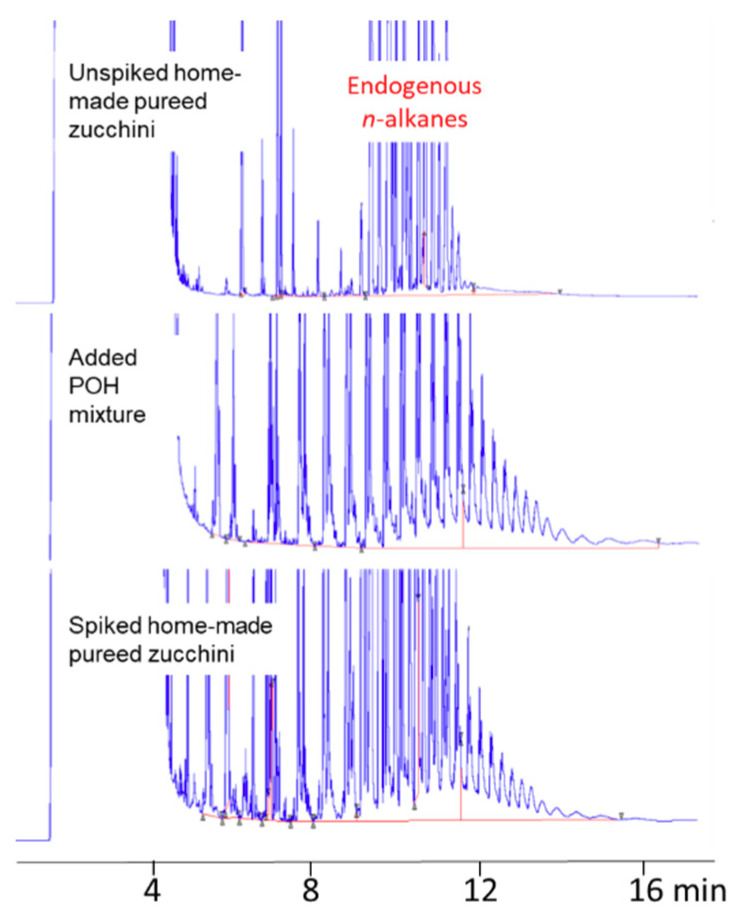
Recovery test.

**Figure 3 foods-09-01365-f003:**
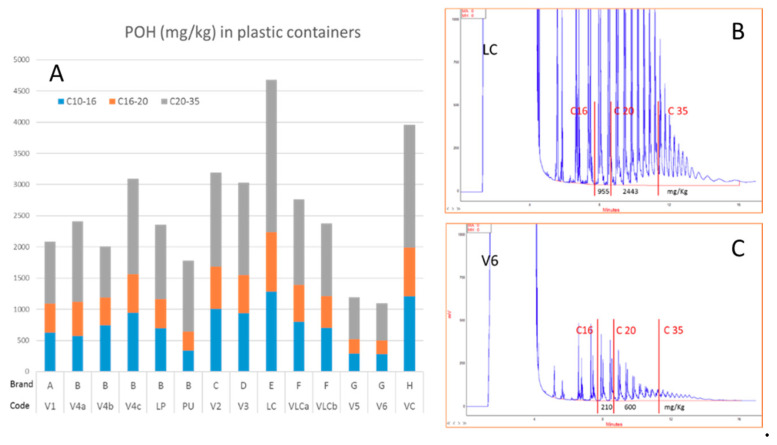
POH content (mg/kg) of the PP containers (**A**) and LC-GC-FID traces of two samples with high (V6 in subgraph (**B**)) and low (LC in subgraph (**C**)) residual POH.

**Figure 4 foods-09-01365-f004:**
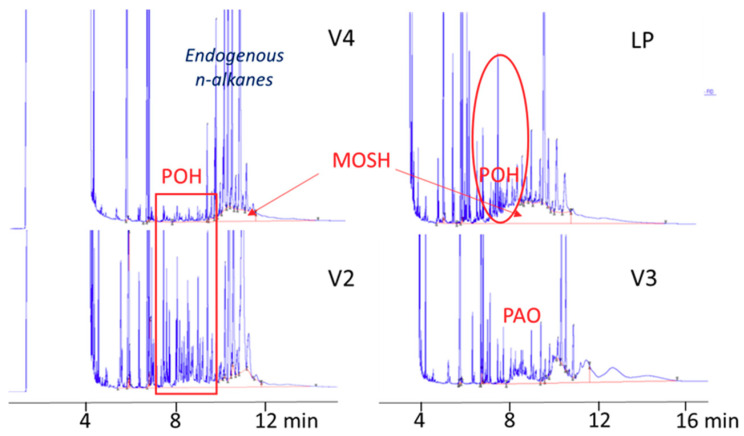
HPLC-GC traces of vegetable soup samples before microwave heating.

**Figure 5 foods-09-01365-f005:**
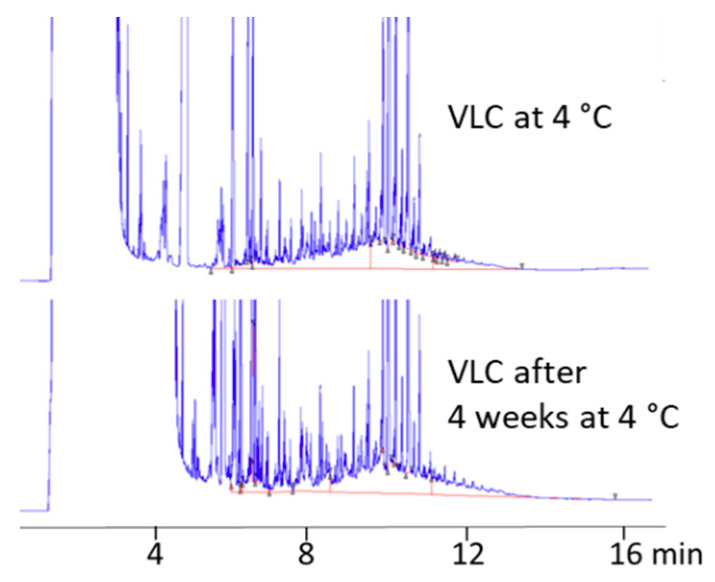
Effect of storage at 4 °C.

**Figure 6 foods-09-01365-f006:**
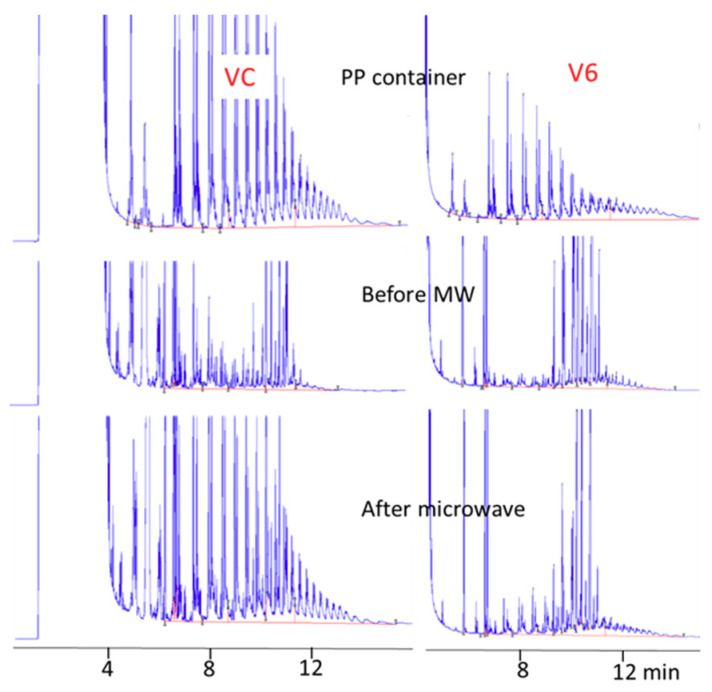
Effect of microwave heating.

**Figure 7 foods-09-01365-f007:**
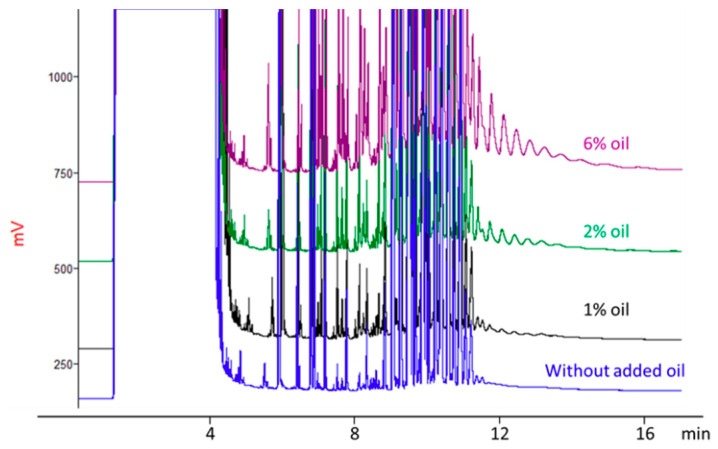
Effect of different oil content on POH migration.

**Figure 8 foods-09-01365-f008:**
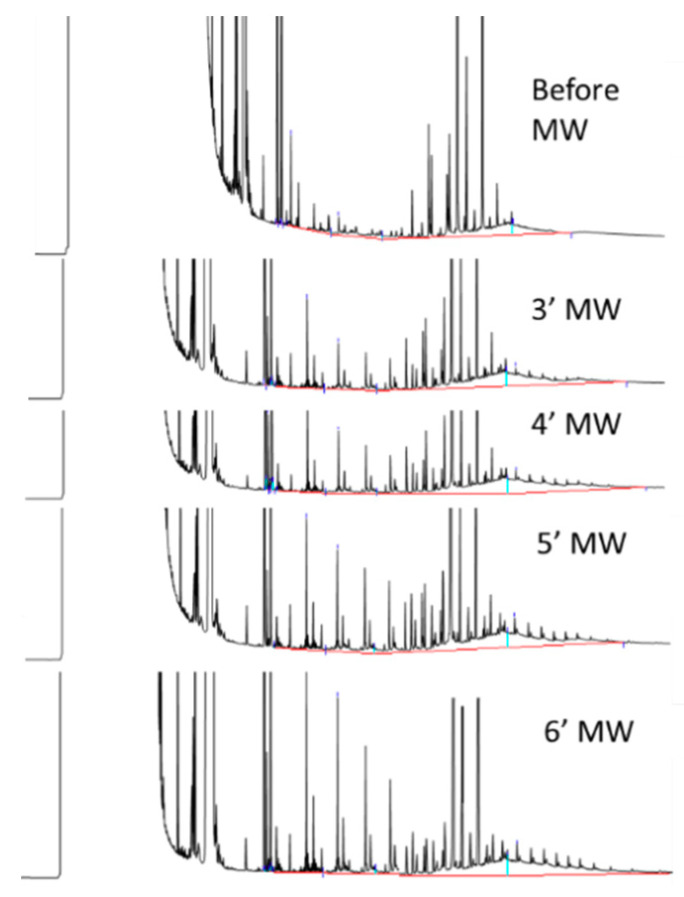
Effect of different heating times on POH migration.

**Table 1 foods-09-01365-t001:** Description of food types.

Sample Code	Ingredients	Brand	Container Weight (g)	Food Weight (g)	Fat (%)	Label Heating Conditions
V1	Mixed vegetables	A	26	620	1.8	900 W for 4’
V2	Mixed vegetables	C	26	600	0.6	750–900 W for 4’
V3	Mixed vegetables	D	26	350	1.0	-
V4	Pumpkin and carrot	B	26	620	2.6	Max power for 3’
V5	Cauliflower and cabbage	G	29	600	1.7	800 W for 4’
V6	Porridge and potatoes	G	29	600	1.9	800 W for 4’
LC	Legumes and cereals	E	32	620	1.0	800 W for 4 min (mix after 3’)
VLC	Vegetables, legumes, and cereals	F	23	600	3.1	800–1000 W for 3’
VC	Vegetables, barley, and spelt	H	26	620	0.5	800–900 W for 4’
LP	Pasta and beans	B	26	600	1.1	Max power for 3’
PU	Potato puree	B	26	450	2.8	Max power for 3’

**Table 2 foods-09-01365-t002:** POH/mineral oil saturated hydrocarbons (MOSH) content (mg/kg) of vegetable soup samples before and after microwave heating and net migration.

Sample Code	Fat (%)	Before MW				After MW				Net POH Migration
		C_10–16_	C_16–20_	C_20–35_	C_10–35_	C_10–16_	C_16–20_	C_20–35_	C_10–35_	C_10–35_
V1	*1.8*	0.3	0.3	0.5	**1.1**	0.4	0.5	3.1	**4.1**	**3.0**
V2	*0.6*	0.7	0.6	1.4	**2.7**	2.4	2.2	4.3	**8.9**	**6.2**
V3	*1.0*	0.4	0.6	2.1	**3.0**	0.3	0.6	1.8	**2.8**	**0.2**
V4	*2.6*	0.1	0.1	0.8	**0.9**	1.2	1.3	4.1	**6.6**	**5.7**
V5	*1.7*	0.1	0.1	0.5	**0.6**	0.1	0.2	0.8	**1.0**	**0.4**
V6	*1.9*	0.1	0.1	0.5	**0.6**	0.1	0.2	0.4	**0.7**	**0.1**
LC	*1.0*	0.3	0.3	0.4	**1.0**	0.3	0.5	0.9	**1.7**	**0.7**
VLC	*3.1*	0.3	0.3	1.0	**1.6**	0.8	0.7	1.4	**3.0**	**1.4**
VC	*0.5*	0.3	0.2	0.3	**0.8**	0.6	0.7	1.2	**2.5**	**1.8**
LP	*1.1*	0.5	0.4	1.3	**2.3**	0.9	0.8	1.6	**3.4**	**1.1**
PU	*2.8*	0.2	0.4	0.2	**0.8**	0.2	0.4	0.4	**0.9**	**0.2**

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
