# Peer review of "Migration of Polypropylene Oligomers into Ready-to-Eat Vegetable Soups"

_foods, 2020, doi:10.3390/foods9101365_

Round 1

Reviewer 1 Report

The authors present a very interesting work on the analysis polypropylene oligomers. The experimental methodology is clearly explained and the results are adequately discussed.

The authors investigate PH residues in PP containers to evaluate the migration of the residues in vegetable soups depending on heating time and the fat amount of the food. The topic has already been investigated (https://doi.org/10.1111/1541-4337.12028) however the authors raise up an interesting point focusing on the but investigation of migration in vegetable soups.

The authors focus on the analysis of a particular fatty food (vegetable soups) under different heating conditions.

Unfortunately, conclusions are vague with the authors stating that ''Detectable POH amounts were already present in food before microwave heating, probably due to hot filling during the production''. They report increase in concentrations, however, after microwave heating proving the findings of their research.

Author Response

The authors present a very interesting work on the analysis polypropylene oligomers. The experimental methodology is clearly explained, and the results are adequately discussed.

The authors investigate PH residues in PP containers to evaluate the migration of the residues in vegetable soups depending on heating time and the fat amount of the food. The topic has already been investigated (https://doi.org/10.1111/1541-4337.12028) however the authors raise up an interesting point focusing on the but investigation of migration in vegetable soups.

The authors focus on the analysis of a particular fatty food (vegetable soups) under different heating conditions.

Unfortunately, conclusions are vague with the authors stating that ''Detectable POH amounts were already present in food before microwave heating, probably due to hot filling during the production''. They report increase in concentrations, however, after microwave heating proving the findings of their research.

The authors thank the reviewer for the comments.

The conclusions have been improved by shifting the comment on POH amounts before microwave heating after reporting MOH/POH amounts before and after microwave heating.

Reviewer 2 Report

Overall, the manuscript is well described. Few minor changes are recommended.

The authors provide some other literature similar to their study in the introduction. However, I would like to encourage expanding this section and provide a more detailed literature survey as this is an area that is not explored much. Maybe in table format. This will be beneficial for readers. 

Line 36: I would give examples of important properties.

Please specify the internal standards in section 2.

Please give details on the quantification calculation.

Figure 1: I would not overlay the chromatograms. It is harder to compare and see.

Specify the GC career gas in the method section.

Author Response

Overall, the manuscript is well described. Few minor changes are recommended.

The authors provide some other literature similar to their study in the introduction. However, I would like to encourage expanding this section and provide a more detailed literature survey as this is an area that is not explored much. Maybe in table format. This will be beneficial for readers.

Two new references have been added. Due to the low number of works focused on the POH subject, and considered that this is not a review, we don’t agree in presenting a table to provide a literature survey.

Line 36: I would give examples of important properties.

Sorry but the reported citation [2] does not report these improved properties, so the text remained unchanged. I suppose that the presence of oligomers may influence polymer viscosity for example.

Please specify the internal standards in section 2

The internal standard solution composition and concentration is reported in section 2 from line 85 to line 89.

Please give details on the quantification calculation.

The text has been modified as follows: “Quantification was based on internal standards based on the following formula:

 whc= (As x m x mis x 1000)/Ais x m

Where:

whc    is the mass fraction of the MOSH/POH in mg/kg

As      is the area of the sample

Ais   is the area of the internal standard peak

mis   is the mass, of the internal standard solution added to the sample, in mg

m    is the mass of the test portion, in g

When MOSH internal standard peaks coeluted with the POH, the MOAH standard or, external standard calculation was used”.

Figure 1: I would not overlay the chromatograms. It is harder to compare and see.

The figure has been modified as suggested by the reviewer

Specify the GC career gas in the method section.

The following statement has been added in the text: “Hydrogen was used as the carrier gas with constant pressure of 60 kPa increased at 90 kPa during the tranfer of the fractions from LC to GC.”